# Persistence of a Wild-Type Virulent *Aeromonas hydrophila* Isolate in Pond Sediments from Commercial Catfish Ponds: A Laboratory Study

**DOI:** 10.3390/vetsci10030236

**Published:** 2023-03-22

**Authors:** James T. Tuttle, Timothy J. Bruce, Hisham A. Abdelrahman, Luke A. Roy, Ian A. E. Butts, Benjamin H. Beck, Anita M. Kelly

**Affiliations:** 1Alabama Fish Farming Center, Greensboro, AL 36744, USA; 2School of Fisheries, Aquaculture, and Aquatic Sciences, Auburn University, Auburn, AL 36849, USA; 3Department of Veterinary Hygiene and Management, Faculty of Veterinary Medicine, Cairo University, Giza 12211, Egypt; 4Aquatic Animal Health Research Unit, US Department of Agriculture, Agricultural Research Service, Auburn, AL 36832, USA

**Keywords:** soil microbiology, persistence, abiotic factors, sediment accumulation, intensive aquaculture, versatility

## Abstract

**Simple Summary:**

In western Alabama, channel and hybrid catfish farmers must constantly deal with disease outbreaks. Virulent *Aeromonas hydrophila* (vAh) is a bacterial pathogen responsible for causing high mortality events in farmed catfish. In recent years, vAh outbreaks have become more chronic and recurring. The main topic of this research project is to determine if vAh can persist within a pond environment. An experimental trial was conducted in the laboratory using glass tanks containing submerged sediments from commercial catfish ponds inoculated with live vAh colonies. Over time, the vAh concentration in the sediments was tracked. It was determined that vAh can persist within the sediments for several weeks. This information will be highly important to catfish producers and will hopefully be used to better our understanding of how and why vAh outbreaks occur.

**Abstract:**

Virulent *Aeromonas hydrophila* (vAh) is a major bacterial pathogen in the U.S. catfish industry and is responsible for large-scale losses within commercial ponds. Administering antibiotic feeds can effectively treat vAh infections, but it is imperative to discern new approaches and better understand the mechanics of infection for this bacterium. As such, the persistence of vAh in pond sediments was determined by conducting laboratory trials using sediment from four commercial catfish ponds. Twelve chambers contained sterilized sediment, vAh isolate ML-09-119, and 8 L of water maintained at 28 °C and were aerated daily. At 1, 2, 4, 6, and 8 days, and every 7th day post-inoculation for 28 days, 1 g of sediment was removed, and vAh colony forming units (CFU) were enumerated on ampicillin dextrin agar. Viable vAh colonies were present in all sediments at all sampling periods. The vAh growth curve peaked (1.33 ± 0.26 × 10^9^ CFU g^−1^) at 96 h post-inoculation. The population plateaued between days 14 and 28. No correlations were found between CFU g^−1^ and physiochemical sediment variables. This study validated the ability of vAh to persist within pond sediments in a laboratory setting. Further research on environmental factors influencing vAh survivability and population dynamics in ponds is needed.

## 1. Introduction

Aquaculture is a rapidly expanding agriculture sector, and the production of farm-raised aquatic organisms is essential to a rapidly growing global population. The commercial production of catfish, which includes channel catfish (*Ictalurus punctatus*) and hybrid catfish [♀ channel catfish (*I. punctatus*) × ♂ blue catfish (*I. furcatus*)], exceeds all other finfish species production in the U.S. In 2021, catfish industry sales were nearly USD 421 million, a 12 percent increase from the previous year [1]. However, these numbers are negatively affected by fish losses due to diseases.

In 2009, a virulent strain of *Aeromonas hydrophila* (vAh) discovered on catfish farms in Alabama and Mississippi became known as a primary pathogen of Motile *Aeromonas* Septicemia (MAS) outbreaks in the U.S. [2]. These outbreaks resulted in high mortalities of farm-raised market-sized catfish and millions of dollars in financial damages [3,4,5,6,7]. *Aeromonas hydrophila* is a Gram-negative, facultative, oligotrophic, ubiquitous anaerobe that causes severe hemorrhaging, exophthalmia, and organ failure in numerous species [8,9,10]. Fish mortalities due to vAh infections can progress rapidly in a pond from a few individuals (5–15%) to the entire pond (up to 100%) in a few days [11], depending on the virulence of the *A. hydrophila* strain [7]. From 2009 to 2021 in Alabama, USA, an estimated 17,064,462 kg of catfish were lost to MAS caused by vAh [12]. From 2015 to 2021, more than 9,500,000 kg of catfish were lost due to vAh, equating to approximately USD 3.4 million annually in foregone sales [13].

Research on vAh has identified how it enters a fish host and affects specific organ systems and what environmental factors influence pathogenesis [14,15]. For example, *A. hydrophila* is efficient in using siderophores to thrive in iron-limited conditions, and this aspect allows for enhanced virulence, as observed in laboratory settings [16]. In addition, the virulence of attenuated vAh isolates can be reduced by removing certain O-antigens [17], and vAh colonies can produce varying concentrations of proteolytic enzymes, adhesins, and toxins depending on their culture status [18]. Thus, components of biofilm formation and secretion systems are also integral to the virulence of vAh and its ability to evade fish defenses [10]. 

Most importantly, there have been multiple studies on how vAh can enter and spread across numerous ponds. Many fish-eating aquatic birds prey on alive, dead, or moribund catfish at commercial catfish facilities [19] and serve as vectors for bacterial pathogens, including vAh [20]. Aquatic birds have high vAh recovery rates, and the primary isolation site is the intestines [21]. Multiple studies demonstrated that vAh is still viable when it passes through the digestive tract of predatory birds. Consequently, the bird feces contain substantial concentrations of vAh capable of infecting fish in numerous ponds, causing severe mortalities on farms that can be miles apart [20,21,22]. Bivalves, aquatic arthropods, and gastropods living in ponds can also harbor vAh, allowing the bacterium to accumulate within aquatic invertebrates [9]. Seining nets can harbor vAh and are most commonly responsible for export to other ponds and farms [2,23]. These studies have furthered our understanding of the intricacies of vAh, but information on the persistence of this bacterium in pond sediments is lacking. 

The bottoms of catfish ponds contain many microorganisms [24] and an accumulation of organic and inorganic materials [25] that may allow pathogenic vAh to persist within this unique environment. The soil, bottom sediments, and biofilms found within catfish production ponds have been known to sequester vAh colonies when water temperatures cool [26]. Notably, Barria et al. [27] reported that cold-response mechanisms are absent in *A. hydrophila*, which would explain the bacterium’s ability to enter a viable but not culturable (VBNC) state at colder temperatures. This VBNC state may be responsible for the appearance of multiple distinct strains of *A. hydrophila*, thereby increasing the genetic heterogeneity of the species [28]. Entering a VBNC state would allow a pathogen such as vAh to decrease the rate of cellular processes and then resume normal functions when environmental conditions improve [29]. Bacterial persistence may also explain the phenomenon of vAh causing chronic and recurring MAS infections [30,31,32,33,34,35]. Understanding the ability of vAh to persist within commercial catfish pond bottoms, and survive over long periods, will improve our knowledge of this harmful bacterial pathogen. The primary goals of this study were to determine if vAh can persist within pond sediments while simultaneously observing how vAh populations change over time and if any physiochemical components of the sediments were correlated with observed vAh population trends. We hypothesized that the vAh populations would exhibit a typical microbial growth curve and that differences in growth curve values would occur between the four sediment types. 

## 2. Materials and Methods

### 2.1. Pilot Trial

Before initiating the full persistence trial (FPT), a pilot-scale study was conducted to confirm the feasibility of the experimental design and to determine if colonies of vAh could be successfully enumerated from an aqueous environment over time. The pilot and FPT sediment samples, water, bacterial inoculum, and aquaria systems were prepared using the methods described below. 

### 2.2. Experimental Design and System Preparation

Approximately 3–4 kg of top layer sediment was collected from six points within four separate production ponds on two farms in Hale County, Alabama, USA. Two ponds from one of the farms (Farm B) had been recently drained. The third pond had just been completely renovated and was to be refilled with water shortly after sample collection. From the fourth pond, which was in production at the sampling time, sediment was collected from the embankments 1 m below the water surface. Both the third and fourth ponds were sourced from Farm A. Sediment samples from each pond were thoroughly mixed to form a single composite sample [36]. Composite samples were then autoclaved at 121 °C, 15 psi, for three 1 h intervals [37,38] using a Market Forge STM-E Sterilmatic Analog Sterilizer (Booth Medical Equipment, Alexander, AR, USA). Once each composite sample was thoroughly autoclaved, each sediment type was quality tested to ensure sterility. The sterilization of soil can increase the extractability of nitrogen, sulfur, phosphorus, organic matter, and notable metal cations, while soil pH, cation exchange capacity (CEC), and surface area typically remain unaffected [38]. Composite samples (1 g each) were vigorously mixed in 15 mL conical tubes (VWR International, Radnor, PA, USA) with sterile deionized water, mixed, aseptically plated onto tryptic soy agar, and incubated at 28 °C for 120 h. If no microbial colonies formed, then soil sterilization was considered successful. If microbial colonies did grow, then the composite sample would be autoclaved for a fourth 1 h interval and re-tested until sterilization was confirmed. 

Dechlorinated city water (96 L) was divided among five containers and disinfected using a 5% chlorine bleach solution [39], with a contact time of 18 h. The remaining chlorine was blown off with filtered air for a minimum of 36 h. The containers were then topped off with autoclaved city water containing sodium thiosulfate to neutralize any remaining free chlorine. Water from each container was tested using a Hydrion CH-300 test strip to ensure all chlorine was neutralized and that all microbial activity had ceased. Quality tests on the water were performed by aseptically adding 5 mL of test water to 5 mL of tryptic soy broth (TSB) and incubating at 28 °C for a total of 120 h. If the solution remained translucent, then water solutions were deemed sterile. If the broth appeared cloudy, then the water disinfection process would be repeated and retested until the city water was free of chlorine and microbial activity ceased. 

The systems consisted of three 37 L glass aquaria divided into four chambers. The chambers were separated by glass panes held in place with aquarium-safe silicone (Silicone 1 All Purpose, General Electric, Waterford, NY, USA). Once the silicone had cured, leak tests on all chambers were conducted to ensure each test chamber was isolated. Before the start of this trial, all tanks were cleaned first with 70% ethanol, followed by 10% Virkon™ S (Antec International, Pittsburgh, PA, USA), and 70% ethanol for a second time. Therefore, these systems would only contain the prepared sediment, water, and vAh culture. Once the aquaria chambers had sediment, vAh, and water, they were covered in two layers of plastic wrap (GLAD^®^ Cling’n Seal, Oakland, CA, USA) and one layer of Styrofoam insulation board (DOW^®^, Midland, MI, USA). This was to limit potential airborne contaminants from entering the system and better maintain temperatures within the chambers. The preparations for the sediments, water, and aquaria systems were not intended to maintain sterility indefinitely but to create an environment in which the bacterial pathogen of interest would be able to propagate initially without competition from other background microorganisms or external factors. The aquaria were kept in a room with an average temperature of 28.0 ± 0.5 °C maintained throughout the trial. 

### 2.3. Bacterial Culture and Trial Preparation

The bacterial culture and inoculum were prepared following the procedure described by Brandi et al. [37]. The wild-type *A. hydrophila* ML-09-119 was isolated from infected catfish during a MAS pond outbreak in west Alabama [40]. ML-09-119 colonies were revived from cryostock by plating on tryptic soy agar (TSA) and incubated at 28 °C for a minimum of 24 h. Next, a pure colony of vAh was picked and placed in 1 L of TSB and incubated at 28 °C for a minimum of 24 h. Next, the bacterial broth culture was centrifuged at 4000× *g* for 10 min in a 5810 R benchtop centrifuge (Eppendorf North America Inc., Enfield, CT, USA), washed in cold 1× phosphate-buffered saline solution (PBS) with an adjusted pH of 7.4. Bacterial cells were resuspended and adjusted to an optical density of 0.200 ± 0.005 at 550 nm using an Eppendorf Biospectrometer^®^ Basic (Eppendorf North America Inc., Enfield, CT, USA). The resulting inoculum had an average concentration of 1.64 × 10^8^ colony-forming units (CFU) per mL. A randomized block design was used to assign chambers to the sediment types. In each chamber, 20 mL of bacteria inoculum was added to 200 g of sterilized sediment and 500 mL of sterilized dechlorinated city water. The soil amalgam was vigorously mixed with a sterile stainless-steel spatula for 1 min durations every 5 min for 1 h. This would ensure adequate contact time, be conducive to keeping the bacteria primarily in the sediment and provide bacteria with a nutrient-rich substrate to vivify and reasonably maintain the population. After the 1 h mixing period, the water volume within each chamber was increased to a total of 8 L. To simulate the mechanical aeration that takes place within a production pond, a 3.5 cm × 1 cm × 1 cm cuboid Pawfly air stone (at a fixed location within each chamber) would expel air supplied via a Whitewater Silent Air Pump™ v201 (Pentair Aquatic Eco-Systems™, Apopka, FL, USA) for 12 h beginning at 1800 h and stopping at 0600 h the following morning. Sediments were left untouched until the first sampling.

### 2.4. Sampling and Bacterial Enumeration

Sediment in each chamber was collected and bacterial populations were evaluated, with sampling times as follows: 24 h post-inoculation (designated as day 0), 48 h post (day 1), 4 d post (day 3), 6 d post (day 5) and 8 d post (day 7), then every seven days following the fifth sampling. Methods described by Cai et al. [26] were used to extract soil and enumerate live colonies of ML-09-119 for each sample. Approximately 1 g of soil was collected from each chamber using a sterile 10 mL serological pipette, placed in a sterile 15 mL centrifuge tube, and centrifuged for 10 min at 667× *g*. Liquid supernatant was removed and the remaining soil pellet (~1 g) was resuspended entirely in 0.1× PBS, creating a 1:10 mixture, and vortexed until the pellet was homogenized. Next, 250 μL of homogenized soil solution was placed into six wells of the leftmost column of a 96-well plate and serially diluted (10-fold) as described by Chen et al. [41]. Four serial dilutions of six 10 μL replicates were each plated onto ampicillin dextrin agar (ADA) for *Aeromonas* spp. selectivity [42]. Plates were dried and then placed in an incubator at 28 °C. The plates required 16 h of incubation at this temperature, and final counts were recorded utilizing the necessary correction factors to determine the CFU g^−1^ of sediment accurately. On each sampling day, viable colonies of vAh were picked and either cryopreserved in a 50% glycerol stock at −80 °C for a separate study or had genomic DNA (gDNA) extracted for PCR confirmation. Any bacteria not confirmed to be vAh were designated as “unknown” and labeled as such, followed by their respective chamber name and sampling day. 

### 2.5. Pilot and FPT Differences

For the pilot trial, only one glass aquarium, divided into four separate chambers, was used. Once sediment samples were inoculated and water volume was increased to the final 8 L per chamber, the system was kept in a room with an average temperature of 21.0 ± 1.2 °C. During the extraction and enumeration process, serial dilutions of sediment samples were plated on ADA via the spread plate method [43] and incubated at 28 °C for 24 h. Two ADA plates were used for each of the four targeted serial dilutions. 

### 2.6. DNA Extraction and PCR Confirmation 

Once ML-09-119 colonies formed on selective agar, the viable isolates were confirmed via polymerase chain reaction (PCR). gDNA from all bacterial colonies was extracted using the EZNA^®^ Bacterial DNA Kit (Omega Bio-tek Inc., Norcross, GA, USA). The concentration and purity of gDNA were measured using a NanoDrop™ One^C^ spectrophotometer (Thermo Fisher Scientific Inc., Waltham, MA, USA). PCR and thermocycling parameters for vAh typing were conducted using methods described by Rasmussen-Ivey et al. [10]. A 25 μL PCR reaction was constructed using 12.5 μL of Hot-Start Taq Master Mix 2X (Amresco LLC, Solon, OH, USA), 0.5 μL of ML-09-119F and ML-09-119R primers (initial 10 μM stock solution), and 75 ng of template gDNA. Thermal cycling runs were conducted using an Eppendorf Mastercycler^®^ X50s (Eppendorf North America Inc., Enfield, CT, USA) with an initial denaturation of 94 °C for 3 min followed by 35 cycles of 94 °C for 30 s, 58 °C for 30 s, and 72 °C for 1 min, with a final extension at 72 °C for 5 min. Positive and negative controls were run in the thermal cycler with test isolates. Then, 5 μL of PCR product was visualized on a 2.0% agarose gel, stained with GelRed (Biotium Inc., Fremont, CA, USA), in a 1.0× Tris-acetate-EDTA (TAE) running buffer using electrophoresis. PCR product bands were visualized via ultraviolet transillumination using a Gel-Doc-Go imaging system (BioRad Inc., Hercules, CA, USA). To accurately identify unknown bacterial colonies, PCR products of four unknown isolates and primers 63F and 1387R [44] were sent to Eurofins Genomics LLC, for genetic sequencing of the 16S rRNA gene. After nucleotide base-pair results were trimmed and aligned in the Molecular Evolutionary Genetics Analysis (MEGA) software version 11 [45], base-pair sequences were inputted in the NCBI Basic Local Alignment Search Tool (BLAST) database [46].

### 2.7. Sediment and Water Chemical Analysis

Sediment and water parameters from the chambers were measured to be used in later correlation analyses in conjunction with potential trends in the CFU g^−1^. After composite samples were autoclaved, a portion of each sediment type was sent to the Soil Forage and Water Testing Laboratory (Auburn, AL, USA) for alkalinity, soil organic matter, and Mehlich 1 extractable micronutrient concentrations. Alkalinity was measured in mg L^−1^ as the equivalent percentage of calcium carbonate (% CaCO_3_ ppm). Total soil organic matter in mg L^−1^ was determined via loss of ignition. Calcium, potassium, magnesium, phosphorus, copper, iron, manganese, zinc, boron, sodium, and aluminum concentrations in mg L^−1^ were measured via inductively coupled argon plasma spectroscopy. Soil pH was measured using a SensION+ PH3^®^ pH and ORP meter equipped with a 5021T electrode (HACH, Loveland, CO, USA). Cation exchange capacity (CEC) was determined using Visual MINTEQ 3.1 software [47]. To calculate the CEC, values of cation concentrations of each sediment type and measured pH were used as inputs to determine the sum of exchangeable cations each sample can adsorb at their respective pH [48]. All CEC values were reported as milliequivalents per 100 g (meq 100 g^−1^) of the sample.

From each test chamber, 10 mL water samples were collected on days 0, 7, 14, 21, and 28. Alkalinity, hardness, total ammonia-nitrogen, nitrite, nitrate, and phosphorus concentrations (mg L^−1^) were measured using a DR3900 visible spectrophotometer (HACH), and the pH as previously described. Water quality parameters were measured to assess any potential effects of water on bacterial colonies. 

### 2.8. Statistical Analyses

Variances in soil chemistry parameters between farms were assessed using a *t*-test. We compared vAh population (log_10_ CFU g^−1^) changes over time among four sediment types using a two-way repeated measures analysis of variance test, with sediment type used as a random blocking factor. Differences in overall log_10_ CFU g^−1^ between farms were determined using a paired *t*-test. If there were significant differences, post hoc analyses were performed using Tukey’s Studentized Range—HSD. To test correlations between soil chemistry parameters and vAh population (log_10_ CFU g^−1^), data from each soil chemistry variable were analyzed for normality. When bivariate normality was verified, data were analyzed through a Pearson correlation. Results not following this assumption were analyzed through a Spearman’s rank correlation. All multiple testing *p*-values for correlation analyses have been adjusted to control the false discovery rate using the Benjamini–Hochberg procedure [49]. The Shapiro–Wilk test was utilized for the normality analysis of the variables. Statistical significance was set at *p* < 0.05. 

For each farm and the overall study, a bacterial persistence curve (BPC) was created by fitting a smoothing spline (SS) model to vAh population data (log_10_ CFU g^−1^; y-axis) at sampling days (x-axis) as previously described by Hussain et al. [50]. To ensure the compromise between the smoothness of the function and the lack of fit, the selection of the smoothing parameter (λ) was based on the restricted maximum likelihood (REML) method [51]. The fitted SS models were used to predict the vAh population using an x-axis scale from 0–28 d with an interval of 0.001. For each BPC, 95% confidence intervals (95% CI) of predicted vAh population curves were created via bootstrapping [52] implemented in the boot package (version 1.3-28) [53]. Data were resampled with replacement 1000 times, with the SS model re-fitted to these data each time. The 95% CIs was determined from the 2.5 and 97.5th percentiles. For BPC estimates, we considered descriptors to differ significantly between farms if their 95% CIs did not overlap. The G*Power 3.1.9.4 was used for sample size calculations [54]. All BPC analyses were performed using R software (version 4.1.1) [55]. All other statistical analyses were performed with SAS^®^ version 9.4 [56]. All figures were plotted using SigmaPlot version 14.5 (Systat Software Inc., San Jose, CA, USA). All data were presented as the mean ± standard error of the mean (*SE*).

## 3. Results

### 3.1. Pilot Trial

The duration of the pilot trial encompassed a total of 113 days, with colonies of vAh present from day 0 to the final sampling day. Across all four sediment types, there is a significant relationship between the population (CFU g^−1^) of vAh and time (Figure 1). After data were log-transformed, there was no difference in population among sampling days from day 58 to day 113. All pairwise comparisons among sampling days from day 58 to day 113 were not statistically different (*p* > 0.05). The population of vAh increased during the first seven days, followed by a moderate decline, then a plateauing event.

Along with colonies of vAh being produced, colonies of unknown bacteria began appearing on the selective ADA 13 days post-inoculation. These novel bacterial colonies were phenotypically and structurally different from the ML-09-119 colonies, which had previously been solitary on the selective media (Figure 2). After following the DNA isolation and thermal cycling procedures described above, PCR product banding displayed distinct differences between the presumed ML-09-119 isolates and these new unknown bacterial colonies (Figure 3).

The NCBI BLAST database indicated that the four unknown bacterial isolates were revealed to be *Pseudomonas tohonis*, *P. alcaligenes*, *P. taiwanensis*, and *Pseudomonas* spp., with percent identifications of 98.34, 98.97, 97.77, and 99.52, respectively. The results of the pilot persistence trial validated the experimental design as a method for enumerating vAh from aquatic sediments. For the FPT, water parameters and specific soil chemical properties were measured and used as environmental descriptors and components for later correlation analyses.

### 3.2. Full Persistence Trial

All representative vAh colonies counted, from all sampling days when such colonies were present, had their respective DNA extracted and were confirmed via PCR methods. In one of the sediment types from Farm A, the unknown bacterial colonies began to appear on the selective ADA media 48 h post-inoculation. By the third sampling day (96 h post-inoculation), unknown colonies were present in all test chambers. The exact identities of most unknown isolates remain unconfirmed at this time. However, they are most likely *Pseudomonas* spp. or a closely related bacterial species based on the previous 16S rRNA sequencing results of the four unknown colonies. 

Across all 12 test chambers, populations of vAh initially experienced a rapid increase, followed by a moderate decline and plateauing pattern (Figure 4). Collectively from chambers containing samples from Farm A and Farm B, population numbers on day seven were higher than those on day zero (*t*_71_ = 7.54, *p* < 0.0001). On day 14, vAh populations decreased and were not different from the population on day zero (*t*_72_ = 0.99, *p* = 0.9739). On days 21 and 28 post-inoculation, average vAh populations were not different (*t*_71_ = 2.39, *p* = 0.2622).

When comparing the sediment types from the two farms, the number of colonies of vAh in Farm B sediment was higher than the vAh colonies in Farm A sediments (Figure 5). On day 14, one of the sediment types from Farm A had data points missing due to a much more drastic reduction in the vAh population in that specific sediment than was anticipated (<10^6^ CFU g^−1^). To avoid any further instances of missing data, serial dilutions from days 21 and 28 were decreased by a power of 10.

When visualizing the raw values of CFU g^−1^ compared to the log_10_ transformed CFU g^−1^ values on the smoothing splines encompassed by 95% CIs (Figure 6), there is a difference in breadths at 90% of peak raw CFU g^−1^ values between Farm A and Farm B sediments (Table 1). Once data was transformed, there were no significant differences in any population peaks, 90% breadths, or 80% breadths.

Water quality parameters did not noticeably fluctuate throughout the FPT (Table 2). After each composite sediment sample was autoclaved, there were no physical and chemical differences in sediment parameters between the sterilized and non-sterilized samples (data not shown). The water in test chambers containing Farm A sediments was not significantly different from that in test chambers containing Farm B sediments. There was no difference in CEC and calcium (Ca^2+^) concentrations between Farm A and Farm B sediments (Figure 7). The boron concentration (B) in all sediment types was below 0.1 mg L^−1^. In addition to CEC and Ca^2+^ concentrations, there were no differences in any physiochemical sediment parameters between Farm A and Farm B sediments (Table 3).

Due to the small sample size of sediment measurements (two ponds per farm, two farms), a power analysis was conducted to determine the number of values (*n*) per physiochemical parameter required to reveal statistically significant differences between farms (Table 3). The small sample size of sediment chemical and physical properties also resulted in all measured sediment parameters exhibiting no correlation to CFU g^−1^ of vAh present (Table 4). The power analysis revealed the sample size required to determine the statistically significant correlations between vAh populations and each of the sediment physiochemical parameters (Table 4). 

## 4. Discussion

Persistence trials carried out in this study demonstrated that vAh can survive in a submerged soil environment in a laboratory setting. The population curve formed in the pilot and FPT followed similar trajectories with an initial rapid growth phase, followed by a steady population decline, concluding in the population of vAh seemingly plateauing within the sediments. Interestingly, the observed population trajectories in CFU g^−1^ between the two vAh trials are worth noting, considering there were only 4 sediment chambers sampled in the pilot trial and 12 total sediment chambers sampled in the FPT. The final log_10_ population average on day 113 of the pilot trial was approximately 5.3 CFU g^−1^. In the FPT, vAh populations in Farms A and B sediments reached average log_10_ values of 5.2 CFU g^−1^ and 6.4 CFU g^−1^, respectively, after 28 days. These trends in bacterial growth curves are consistent with the findings of other studies on the effect of temperature on the growth dynamics of *A. hydrophila* isolates. While the optimum growth temperature of *Aeromonas* spp. is 20–35 °C, certain strains of *A. hydrophila* can experience positive growth rates from as cold as 0 °C to as warm as 55 °C [57]. Park and Ha [58] also reported that *A. hydrophila* is psychrotrophic due to its ability to continue population increases on squid (*Sepioteuthis sepioidea*) even at 5 °C. Similarly, evaluating the growth rates of *A. hydrophila* on raw tuna (*Thunnus orientalis*), Kim et al. [59] reported an increase in CFU of the target bacterial species over a 168 h period between 8–15 °C. Storage temperatures are critical environmental factors when developing predictive bacterial growth models and reporting the effects of temperature on specific growth rates of *A. hydrophila* [57,58,59]. Future persistence trials at temperatures other than 28 °C and 21 °C would need to be conducted to determine if the population trends of vAh observed in this study are similar or if they die off. Additionally, studies examining cyclical water temperature regimes would more readily reflect natural water temperature fluctuations in commercial aquaculture ponds. 

Other environmental factors that did not correlate with bacterial growth in this study should also be considered for future studies. For example, adding one physiochemical component of sediment may reveal correlations under controlled conditions where sediment is absent. On the other hand, adding two or more of the physiochemical components of soil may demonstrate synergism, antagonism, or no effect. Replicating these trials in a commercial pond would be unrealistic. 

In this study, the wild-type vAh isolate ML-09-119 was selected primarily due to the extensive research already conducted on this strain. Still, numerous other strains of vAh have been documented after the first pathotype was identified multiple decades ago [60]. Isolates originating from Alabama, Mississippi, and Chinese provinces share a common ancestor [6]; however, there is a higher degree of genetic heterogeneity among MS vAh isolates, and there are distinct subclades among U.S. and Chinese strains [10]. Due to the strain diversity of vAh, isolates from different geographical regions may exhibit different persistence behaviors.

During the pilot and FPT, once the unknown bacterial colonies began appearing on the ampicillin agar plates, they were detected in all chambers until the trials were concluded. Population trends of the background, putative *Pseudomonas* spp. during the pilot and FPT were not within the scope of this study; however, these bacterial species did not ultimately outcompete the vAh population. The versatility of vAh to withstand environmental and ecological difficulties could explain this. A diverse array of bacteria can persist in the soils or sediments over long periods [37]. In addition, there are adaptations specific to *A. hydrophila* that improve persistence and survivability in harsh aquatic environments. The adaptations include alternative sigma factors, two-component regulatory systems, chaperones, DNA-damage repair pathways, acid resistance systems, and starvation and antibiotic response mechanisms [28,61,62]. *A. hydrophila* can also metabolize a wide variety of carbohydrates [63], specifically chitin, a major component in the aquatic ecosystem [64,65,66]. Through experimental trials, Zhang et al. [66] reported that vAh isolate ML-10-51K could rapidly proliferate using colloidal chitin and chitin flakes as a sole carbon source at the same rate as if it were supplied glucose. Additionally, virulence factors expressed by establishing biofilm colonies of vAh [18] may allow for more specific niche partitioning [67] between vAh and closely related bacteria such as *Pseudomonas* and non-virulent *Aeromonas* spp.

All populations of microorganisms cultured in a closed or batch system exhibit a consistently shaped growth curve consisting of a lag phase, exponential or logarithmic growth phase, stationary phase, and finally, a death phase [29]. However, in any system, complete cell death of a bacterial population is not likely to occur within a short period, as microbial populations exhibit dynamic patterns of ecological succession when environmental changes occur [29,68,69]. In comprehensive studies, researchers noted that successional trends in bacterial populations are dynamic and challenging to predict. However, the taxonomic and functional bacterial community diversities are highest in the initial years of development and then gradually decrease as an ecosystem becomes more developed [68,69]. Understanding ecological interactions of bacteria, such as interspecific and intraspecific competition, and succession between bacterial communities in commercial catfish pond bottoms would improve the understanding of the mechanisms influencing vAh populations.

This study failed to identify which environmental factors affected the duration of growth, decline, and plateau periods of vAh. The power analysis conducted estimated a total of between 269 and 7023 samples would be required for testing to determine differences at a high-power level. Unfortunately, that sample number is cost-prohibitive. Although the correlation analyses were weak in this study, increasing sample sizes could reduce variation and perhaps establish better correlative values. 

Soil microbiome populations are influenced and controlled by multiple abiotic and environmental factors, and these interrelating abiotic factors complicate analyses of specific influences on individual microbial species [70]. Few studies have analyzed relationships between bacteria and specific soil chemical properties in *Aeromonas* spp. For example, *A. veronii* can volatilize selenium (Se) and produce hazardous chemical products such as dimethyl disulfide, methyl selenol, dimethyl selenosulfide, and dimethyl diselenide; however, the rate of volatilization is dependent on pH and salinity of the environment [71]. Awan et al. [28] noted environmental factors, including temperature, pH, surface hydrophobicity, magnesium transport, flagella expression, chemotaxis nutrient limitation, oxygen deprivation, and quorum sensing (QS) could influence the attachment and establishment of colony-forming biofilms of *A. hydrophila*. Isolates of *A. sobria* and another *Aeromonas* spp. collected from mining site soil in Nigeria displayed strong tolerance to lead (Pb), cadmium (Cd), copper (Cu), and chromium (Cr) at concentrations ≥ 6 mg L^−1^ [72]. Cai et al. [26] noted that aeromonad populations positively correlate with temperature, nitrogen concentration, organic carbon load, and primary productivity. Further research is necessary to determine the relationships between the soil chemistry of catfish ponds and pathogenic bacteria such as vAh.

## 5. Conclusionss

Virulent *A. hydrophila* can persist within pond sediments of commercial catfish ponds. This ability to survive within pond sediments may allow vAh populations the opportunity for horizontal genetic transfer (HGT) [73], which can result in more virulent and robust strains of vAh. In addition, with vAh populations persisting in the sediments over long periods, these long-lasting bacterial populations may be more capable of developing antimicrobial resistance via HGT [74]. Therefore, future research projects focusing on understanding the mechanisms and virulence factors that enable vAh to persist are paramount. In addition, further research is needed to determine which sediment physiochemical parameters influence vAh persistence and if other bacterial strains present in pond bottom sediment can outcompete vAh over time.

## Figures and Tables

**Figure 1 vetsci-10-00236-f001:**
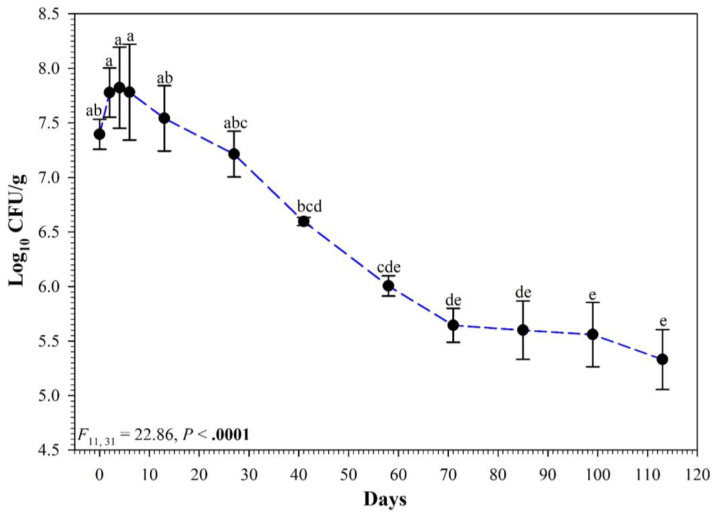
Persistence of virulent *Aeromonas hydrophilia* population (log_10_ CFU g^−1^) over a period of 113 days in soil samples collected from two farms for the pilot trial. Each symbol indicates the mean of four soil samples (two ponds per farm), and error bars around the symbol represent the standard error of the mean. Symbols with different lowercase letters are significantly different at *p* < 0.05.

**Figure 2 vetsci-10-00236-f002:**
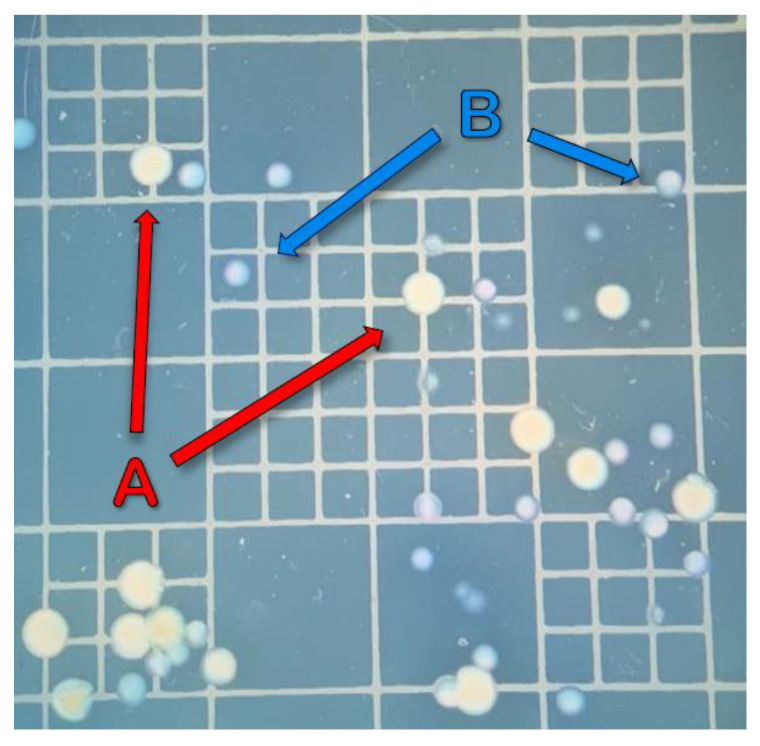
Colony growth of virulent *Aeromonas hydrophilia* isolate ML-09-119 (**A**), and unknown bacterial isolate (**B**) on ampicillin dextrin agar. Sampling day 13 during pilot trial.

**Figure 3 vetsci-10-00236-f003:**
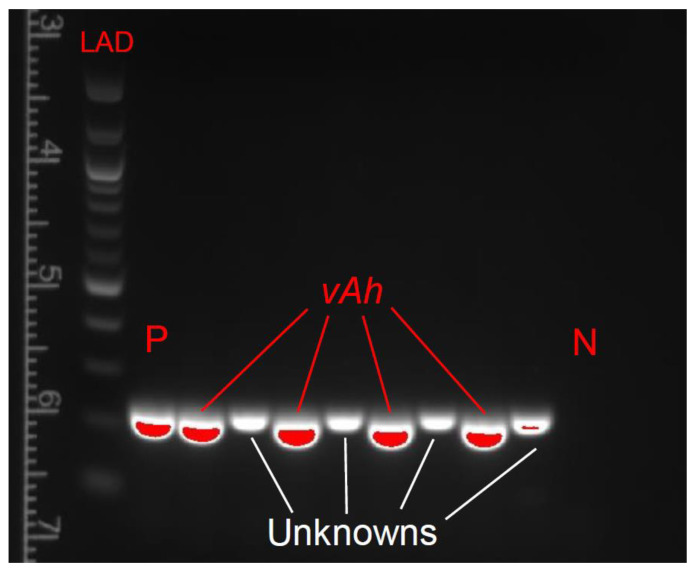
PCR product banding of virulent *Aeromonas hydrophilia* (vAh) and unknown bacterial isolates collected from pilot trial day 41. Positive and negative controls and the DNA ladder were labeled as P, N, and LAD, respectively. Agarose gel (2%) image visualized on an ultraviolet transillumination after. Isolates of vAh formed product bands at 246 bp.

**Figure 4 vetsci-10-00236-f004:**
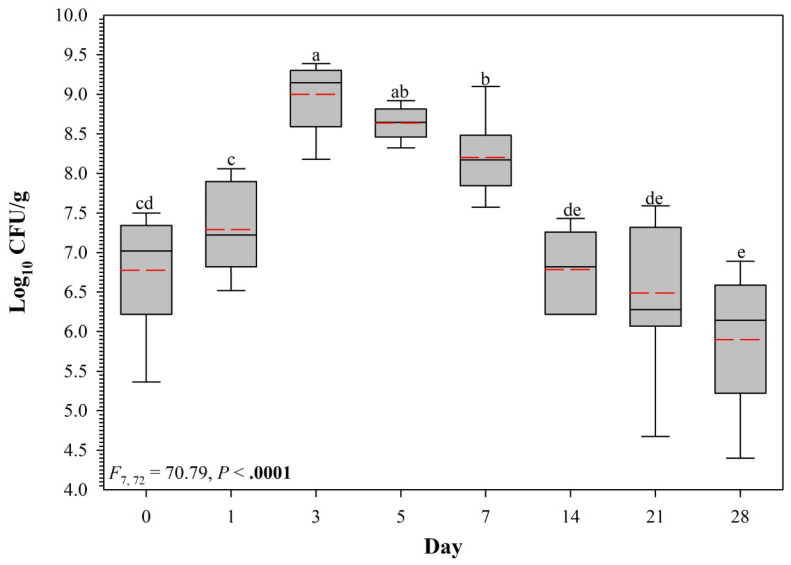
Persistence of virulent Aeromonas hydrophilia ML-09-119 population (log_10_ CFU g^−1^) in soil samples collected from 12 study chambers (2 farms × 2 ponds/farm × 3 replicate tanks per pond). Within each box plot, solid black horizontal line indicates the median, dashed red horizontal line indicates the mean. Box plots with different lowercase letters are significantly different at *p* < 0.05.

**Figure 5 vetsci-10-00236-f005:**
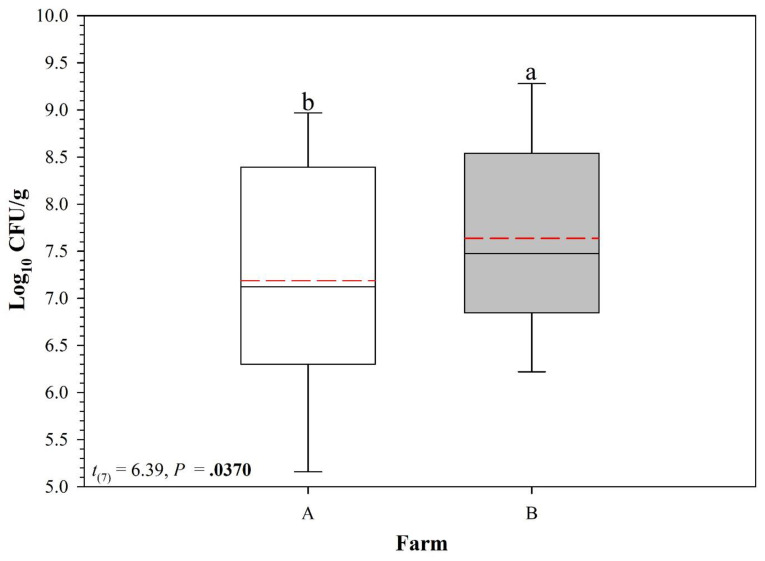
Comparison of virulent Aeromonas hydrophilia population (log_10_ CFU g^−1^) in soil samples collected from 2 farms (2 ponds/farm × 3 replicate tanks per pond), Within each box plot, solid black horizontal line indicates the median, dashed red horizontal line indicates the mean. Box plots with different lowercase letters are significantly different at *p* < 0.05.

**Figure 6 vetsci-10-00236-f006:**
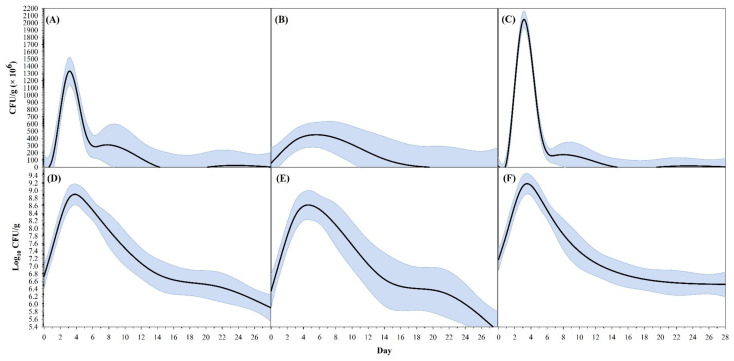
Relationship between virulent *Aeromonas hydrophilia* population in soil (CFU g^−1^: (**A**–**C**); log_10_ CFU g^−1^: (**D**–**F**)) and time (days) using a smoothing spline (SS) model and 95% confidence intervals (gray shadow). Left figures (**A**,**D**) represent all samples; middle figures (**B**,**E**) represent farm A; right figures (**C**,**F**) represent Farm B. Estimates of SS model descriptors are summarized in Table 1.

**Figure 7 vetsci-10-00236-f007:**
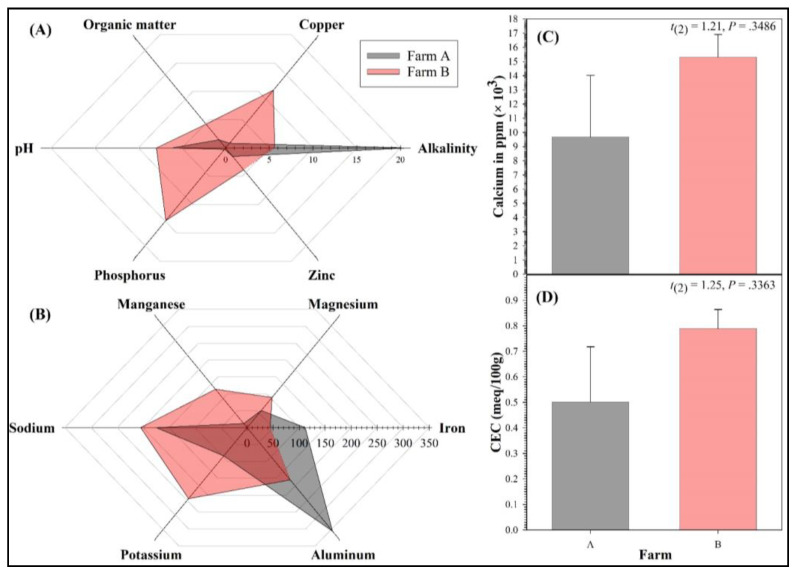
Radar plots (**A**,**B**) and bar charts (**C**,**D**) of sediment chemistry parameters measured in samples collected from 2 farms (2 ponds per farm). Error bars in bar charts represent the standard error of the mean.

**Table 1 vetsci-10-00236-t001:** Mean, standard error (*SE*), and 95% confidence intervals (CI) for descriptors of smoothing spline models presented in Figure 5. Statistical differences in endpoints between farm soil are denoted with an asterisk (*).

Curve Descriptor		Overall		Farm A		Farm B	
		Mean ± *SE*	95% C.I.	Mean ± *SE*	95% C.I.	Mean ± *SE*	95% C.I.
CFU g^−1^	Peak CFU g^−1^ (×10^6^)	1329.8 ± 260.9	818.4–1841.1	449.6 ± 222.0	14.5–884.7	2046.6 ± 1266.6	−435.9–4529.1
	Time (day) at peak CFU g^−1^	3.16 ± 0.19	2.78–3.53	5.58 ± 1.92	1.82–9.34	3.15 ± 0.79	1.61–4.69
	Time (day) at 90% of peak CFU g^−1^—lower	2.64 ± 0.12	2.41–2.88	3.52 ± 1.43	0.72–6.32	2.66 ± 0.79	1.10–4.21
	Time (day) at 90% of Peak CFU g^−1^—upper	3.73 ± 0.32	3.11–4.35	7.97 ± 2.45	3.16–12.77	3.69 ± 0.77	2.18–5.20
	Breadth at 90% of peak CFU g^−1^ *	1.09 ± 0.24	0.61–1.56	4.45 ± 1.47	1.55–7.34	1.03 ± 0.04	0.95–1.11
	Time (day) at 80% of peak CFU g^−1^—lower	2.41 ± 0.10	2.21–2.60	2.82 ± 1.25	0.37–5.27	2.43 ± 0.79	0.88–3.99
	Time (day) at 80% of peak CFU g^−1^—upper	4.00 ± 0.47	3.09–4.92	9.07 ± 2.70	3.77–14.36	3.94 ± 0.76	2.45–5.44
	Breadth at 80% of peak CFU g^−1^ *	1.60 ± 0.44	0.74–2.45	6.25 ± 2.06	2.22–10.28	1.51 ± 0.06	1.40–1.62
	Time (day) at 5% of peak CFU g^−1^—min	0.90 ± 0.19	0.52–1.27	0.00 ± 0.32	−0.62–0.62	1.04 ± 0.71	−0.36–2.43
	Time (day) at 5% of peak CFU g^−1^—max	12.89 ± 1.89	9.17–16.60	17.32 ± 4.46	8.57–26.07	11.22 ± 1.95	7.39–15.05
	Range at 5% of peak CFU g^−1^	11.99 ± 1.89	8.28–15.70	17.32 ± 4.59	8.31–26.32	10.18 ± 1.88	6.50–13.86
Log_10_ CFU g^−1^	Peak log_10_ CFU g^−1^	8.90 ± 0.12	8.67–9.13	8.62 ± 0.12	8.38–8.85	9.18 ± 0.38	8.43–9.93
	Time (day) at peak log_10_ CFU g^−1^	3.80 ± 0.23	3.35–4.25	4.59 ± 0.69	3.24–5.95	3.53 ± 0.79	1.98–5.07
	Time (day) at 90% of peak log_10_ CFU g^−1^—lower	1.71 ± 0.16	1.40–2.01	1.94 ± 0.17	1.61–2.26	1.54 ± 0.84	−0.11–3.19
	Time (day) at 90% of peak log_10_ CFU g^−1^—upper	7.79 ± 0.78	6.26–9.31	9.28 ± 1.30	6.73–11.83	6.63 ± 0.84	4.98–8.28
	Breadth at 90% of peak log_10_ CFU g^−1^	6.08 ± 0.84	4.42–7.73	7.34 ± 1.27	4.85–9.84	5.09 ± 0.61	3.88–6.29
	Time (day) at 80% of peak log_10_ CFU g^−1^—lower	0.54 ± 0.26	0.03–1.04	0.74 ± 0.26	0.23–1.26	0.26 ± 0.88	−1.47–1.98
	Time (day) at 80% of peak log_10_ CFU g^−1^—upper	11.75 ± 1.61	8.59–14.91	12.68 ± 4.09	4.66–20.70	10.23 ± 1.85	6.61–13.85
	Breadth at 80% of peak log_10_ CFU g^−1^	11.22 ± 1.71	7.87–14.56	11.94 ± 4.10	3.89–19.98	9.97 ± 1.83	6.38–13.57

**Table 2 vetsci-10-00236-t002:** Water quality parameters (mean, standard error (*SE*), minimum measurement (min), and maximum measurement (max)) measured in 12 study tanks containing sediment samples (2 farms; 2 ponds per farm; 3 replicate tanks per pond) for 28 d.

Water Quality Parameter	Overall		Farm A		Farm B	
	Mean ± *SE*	Min–Max	Mean ± *SE*	Min–Max	Mean ± *SE*	Min–Max
Total alkalinity (ppm)	120.27 ± 3.53	87–206	111.97 ± 3.65	87–167	128.57 ± 5.71	90–206
Total hardness (ppm)	118.93 ± 4.26	67–199	115.60 ± 5.25	67–190	122.27 ± 6.75	73–199
pH	7.59 ± 0.02	7.3–7.9	7.59 ± 0.03	7.4–7.9	7.59 ± 0.03	7.3–7.9
Phosphate (ppm)	1.18 ± 0.15	0.0–4.0	1.50 ± 0.25	0.0–4.0	0.85 ± 0.15	0.0–2.8
Total ammonia nitrogen (ppm)	0.60 ± 0.10	0.0–3.7	0.33 ± 0.05	0.0–1.3	0.88 ± 0.19	0.0–3.7
Nitrite (ppm)	0.07 ± 0.02	0.0–1.0	0.04 ± 0.02	0.0–0.5	0.11 ± 0.04	0.0–1.0
Nitrate (ppm)	0.25 ± 0.06	0.0–1.0	0.20 ± 0.07	0.0–1.0	0.30 ± 0.09	0.0–1.0

**Table 3 vetsci-10-00236-t003:** Sediment chemistry parameters (mean, standard error (*SE*), minimum measurement (min), and maximum measurement (max)) measured in soil samples collected from 2 farms (2 ponds per farm), test statistics (*t*) and *p*-values of statistical comparison between farms, and the sample size (*n*) required to reveal statistically significant differences between farms.

Soil Parameter	Overall		Farm A		Farm B		Farm A versus Farm B		
	Mean ± SE	Min–Max	Mean ± SE	Min–Max	Mean ± SE	Min–Max	*t* _(2)_	*p*-Value	*n*/Farm
Alkalinity (% CaCO_3_ Equivalence)	13.23 ± 9.36	0.70–41.00	20.85 ± 20.15	0.70–41.00	5.60 ± 1.70	3.90–7.30	0.75	0.5295	29
Aluminum (ppm)	246.80 ± 141.25	8.20–648.00	328.10 ± 319.90	8.20–648.00	165.50 ± 64.50	101.00–230.00	0.50	0.6677	65
Calcium (×1000 ppm)	12.49 ± 2.50	5.32–16.91	9.67 ± 4.36	5.32–14.03	15.31 ± 1.60	13.71–16.91	1.21	0.3486	12
CEC (meq/100 g)	0.65 ± 0.13	0.29–0.86	0.50 ± 0.22	0.29–0.72	0.79 ± 0.08	0.71–0.86	1.25	0.3363	12
Copper (ppm)	5.90 ± 4.71	0.60–20.00	0.90 ± 0.30	0.60–1.20	10.90 ± 9.10	1.80–20.00	1.10	0.4699	15
Iron (ppm)	78.00 ± 36.40	32.00–186.00	111.50 ± 74.50	37.00–186.00	44.50 ± 12.50	32.00–57.00	0.89	0.4687	21
Magnesium (ppm)	75.00 ± 24.82	30.00–143.00	54.50 ± 24.50	30.00–79.00	95.50 ± 47.50	48.00–143.00	0.77	0.5232	28
Manganese (ppm)	67.25 ± 33.30	10.00–149.00	13.00 ± 3.00	10.00–16.00	121.50 ± 27.50	94.00–149.00	3.92	0.0593	3
Organic Matter (%)	3.03 ± 0.99	1.20–5.70	1.55 ± 0.35	1.20–1.90	4.50 ± 1.20	3.30–5.70	2.36	0.1422	5
pH	7.03 ± 0.69	5.03–7.94	6.13 ± 1.10	5.03–7.23	7.93 ± 0.01	7.92–7.94	1.64	0.3492	7
Phosphorus (ppm)	6.98 ± 5.72	0.10–24.00	0.25 ± 0.15	0.10–0.40	13.70 ± 10.30	3.40–24.00	1.31	0.4160	11
Potassium (ppm)	157.25 ± 64.12	50.00–343.00	89.50 ± 39.50	50.00–129.00	225.00 ± 118.00	107.00–343.00	1.09	0.3899	15
Sodium (ppm)	189.50 ± 22.63	151.00–250.00	174.50 ± 23.50	151.00–198.00	204.50 ± 45.50	159.00–250.00	0.59	0.6173	47
Zinc (ppm)	2.85 ± 1.61	0.70–7.50	1.60 ± 0.90	0.70–2.50	4.10 ± 3.40	0.70–7.50	0.71	0.5509	33

**Table 4 vetsci-10-00236-t004:** Results from correlation analysis tests between log_10_ CFU g^−1^ of vAh and sediment chemistry variables. Based on bivariate normality testing, test used was Spearman’s rank correlation (coefficient = *ρ*). All raw *p*-values were adjusted to control the false discovery rate (FDR) using the Benjamini–Hochberg procedure. Significant results at *p* < 0.05. The sample size (*n*) required to reveal statistically significant correlations.

Variable	*ρ*	*p*-Value		*n*/Farm
		Raw	FDR	
Alkalinity (% CaCO_3_ equivalence)	0.13	0.2318	0.3327	487
Aluminum (ppm)	−0.13	0.2318	0.3327	487
Calcium (ppm)	0.12	0.2570	0.3327	542
CEC (meq 100 g^−1^)	0.12	0.2570	0.3327	542
Copper (ppm)	0.19	0.0740	0.3327	219
Iron (ppm)	−0.12	0.2570	0.3327	542
Magnesium (ppm)	0.17	0.1075	0.3327	269
Manganese (ppm)	0.11	0.3089	0.3327	672
Organic matter (%)	0.11	0.3089	0.3327	672
pH	0.16	0.1219	0.3327	291
Phosphorus (ppm)	0.11	0.3089	0.3327	672
Potassium (ppm)	0.17	0.1075	0.3327	269
Sodium (ppm)	−0.03	0.7657	0.7657	7823
Zinc (ppm)	0.13	0.2272	0.3327	478

## Data Availability

All data from this study are available from the corresponding author upon reasonable request.

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
