# Peer review of "Persistence of a Wild-Type Virulent Aeromonas hydrophila Isolate in Pond Sediments from Commercial Catfish Ponds: A Laboratory Study"

_vetsci, 2023, doi:10.3390/vetsci10030236_

Round 1

Reviewer 1 Report

General comments:

The manuscript vetsci-2297021, titled "Persistence of a Wild-Type Virulent Aeromonas hydrophila Isolate in Pond Sediments from Commercial Catfish Ponds: A Laboratory Study" submitted to Veterinary Sciences by Tuttle et al. investigated the persistence of virulent Aeromonas hydrophila (vAh) within catfish pond sediments. The experiments involved inoculating vAh colonies into glass tanks containing sterile commercial catfish pond sediments and tracking vAh concentration over time. Results revealed that vAh can persist in the sediments for several weeks, providing important information for catfish producers to better understand and prevent vAh outbreaks. Generally, well-designed experiments and appropriate methodology are used for the research. The manuscript is well written and targets an important fish pathogen. This reviewer has made only minor comments below that can potentially improve the manuscript.

Specific comments:

1)      Lines 118-119: Composite samples were autoclaved for three (or more) 1-h intervals. Autoclaving soil has been shown to influence soil chemical properties (Wolf and Skipper, Chapter 3 - Soil Sterilization, 1994). Were there any differences between sterile vs control (non-sterile) sediment?

2)      Line 130-131: “Dechlorinated city water was sterilized using a 5% chlorine bleach solution”, suggest rephrasing. Use disinfected instead of sterilized. Chlorine bleach does not sterilize because it does not kill spores that would germinate afterwards.

3)      Line 131-132: Was the air used to blow off chlorine sterile/filtered? Air can be a source of contamination.

4)      Lines 286-288: Delete unnecessary text.

5)      Figure 3: All PCR amplicons shown have the same band size, the difference is in band intensity. This is a PCR with specific primers for the vAh strain (according to Rasmussen-Ivey reference). There should be no amplification with the unknown (non-vAh) isolates. You might have a contaminated stock/culture at the time of DNA extraction.

6)      Line 325: “All colonies….”, suggest rephrasing to representative colonies were confirmed because it is impossible to confirm every single colony in a study like this one.

7)      Line 351-353: Provide a proper figure legend to Figure 5.

8)      It was a pleasure to review this manuscript.

Author Response

1)  Lines 118-119: Composite samples were autoclaved for three (or more) 1-h intervals. Autoclaving soil has been shown to influence soil chemical properties (Wolf and Skipper, Chapter 3 - Soil Sterilization, 1994). Were there any differences between sterile vs control (non-sterile) sediment?

There were no differences between the sterile and non-sterile sediment samples. Reporting of the physiochemical parameters of non-sterile sediment samples was beyond the scope of this study.

2) Line 130-131: “Dechlorinated city water was sterilized using a 5% chlorine bleach solution”, suggest rephrasing. Use disinfected instead of sterilized. Chlorine bleach does not sterilize because it does not kill spores that would germinate afterwards.

Changes made as requested.

3) Line 131-132: Was the air used to blow off chlorine sterile/filtered? Air can be a source of contamination.

The air used to blow off the chloride was filtered. Changes made as requested

4) Lines 286-288: Delete unnecessary text.

Correction made as requested

5) Figure 3: All PCR amplicons shown have the same band size, the difference is in band intensity. This is a PCR with specific primers for the vAh strain (according to Rasmussen-Ivey reference). There should be no amplification with the unknown (non-vAh) isolates. You might have a contaminated stock/culture at the time of DNA extraction.

While the product banding between the vAh isolates and the unknowns appears very similar in Figure 3, There was a perceived difference in product band size and intensity between the unknown isolate PCR products and the vAh PCR products concerning their location next to the DNA ladder. These minor differences are what prompted the need for 16s rRNA sequencing. Also, in the Rasmussen-Ivey publication, they mention that the ML09-119F and ML09-119R primers amplify a lineage-specific amplicon that is not specific to only the wild-type ML-09-119 isolate, and they were able to identify a total of 20 different isolates using those specific primers (See: Results; Identifying new vAh isolates).

Contamination of the unknown bacterial isolates is a possibility. However, the PCR confirmation was conducted on isolates collected 13 days post sediment inoculation. It is also a possibility that while exposed to that environment, there was some horizontal genetic transfer that would allow the unknown isolates to produce some product banding with the ML-09-119 specific primers. 

6) Line 325: “All colonies….”, suggest rephrasing to representative colonies were confirmed because it is impossible to confirm every single colony in a study like this one.

Changes made as requested.

7) Line 351-353: Provide a proper figure legend to Figure 5.

Correction made as requested.

8) It was a pleasure to review this manuscript.

Thank you for your time and consideration. Your constructive comments during this review process are greatly appreciated.

Reviewer 2 Report

The paper is well written, innovative and provides a better understanding of the transmission of infections caused by the virulent Aeromonas hydrophila bacterial pathogen. The experiment is well set up and done, the methods as well. The results are presented adequately, with statistical processing done. In the discussion, the obtained results were explained, adequate and current literature was used, and the conclusion is in accordance with the key findings of the work itself, and it was emphasized what should still be processed in subsequent research.

Based on everything, I believe that the work is important for a better understanding of the issues it deals with and that it can be published without additional changes.

Author Response

Thank you very much for your time and consideration. Your feedback during this review process is greatly appreciated.